# DEEP GRAPH TRANSLATION

## ABSTRACT

Deep graph generation models have achieved great successes recently, among which however, are typically unconditioned generative models that have no control over the target graphs given an input graph. In this paper, we propose a novel Graph-Translation-Generative-Adversarial-Networks (GT-GAN) that transforms the input graphs into their target output graphs. GT-GAN consists of a graph translator equipped with innovative graph convolution and deconvolution layers to learn the translation mapping considering both global and local features. A new conditional graph discriminator is proposed to classify the target graphs by conditioning on input graphs while training. Extensive experiments on multiple synthetic and real-world datasets demonstrate that our proposed GT-GAN significantly outperforms other baseline methods in terms of both effectiveness and scalability. For instance, GT-GAN performs at least 10 and 15 time faster than GraphRNN and RandomVAE, respectively, when the size of the graph is around 50.

## 1 INTRODUCTION

In recent years, deep learning on graphs has seen a surge of interests, especially for graph representation and recognition tasks such as node-level classification (Li et al., 2016; Kipf & Welling, 2017; Veličković et al., 2017; Gilmer et al., 2017; Hamilton et al., 2017) and graph-level classification (Niepert et al., 2016; Atwood et al., 2016; Wu et al., 2017). Because of the successes in graph neural networks, researchers have recently started to explore the use of deep generative models for graph synthesis on practical applications such as designing new chemical molecular structures (Simonovsky & Komodakis, 2018; You et al., 2018). This has led to many of the recent advances in deep graph generative models where some of these approaches are domain dependent models (Kusner et al., 2017; Dai et al., 2018) for generating graphs with physical constrains, while some others consider the generation of generic graphs (Li et al., 2018; Samanta et al., 2018; Jin et al., 2018a).

However, there are two main drawbacks of existing deep graph generative models. First, one significant limitation of the previous approaches is that most of these models are only suitable for small graphs with 40 or fewer nodes, which is mainly due to their one-node-per-step generation manner. More importantly, most of the existing graph generation models are unconditioned and thus ignore rich input graph information for generating a new graph. In many applications, it is crucial to guide the graph generation process by conditioning on an input graph, which can be cast as a graph translation learning problem – translating the input graph to the output graph.

One straightforward way is to build a translation system by using a graph encoder-decoder architecture. However, there are several challenges for this type of approaches: 1) *how to learn one-to-more mapping between the input graph and the target graphs*. Different from the plain graph generation problem, a conditional graph synthesis task is to learn a distribution of target graphs conditioning on the input graph, which aims to capture the underlying implicit properties of the graphs, such as their scale-free characteristic. 2) *how to jointly learn both local and global information for translation*. One needs to not only learn the translation mapping in the local information (i.e. neighborhood pattern of each node), but also in the global property of the whole graph (e.g.,node degree distribution or graph density).

To address the aforementioned challenges, we present a novel neural network architecture – Graph-Translation-Generative-Adversarial-Nets (GT-GAN). We first propose a conditional graph GAN architecture that consists of an encoder-decoder translator and a conditional graph discriminator to learn the one-to-more mapping (a conditional distribution) for graph translation. To jointly embed

the local and global information, we present a novel graph encoder including both the edge and the node convolution layers. In addition, we further propose a novel graph U-net with graph skips and dedicated graph deconvolution layers including both the edge and the node deconvolution layers. Finally, GT-GAN is scalable with at most quadratic computation and memory consumption in terms of the number of nodes in a graph, making it suitable for at least modest-scale graphs (with hundreds of nodes, compared to the tens of nodes in most of existing graph generative models).

We highlight our main contributions as follows:

- We develop a generic framework GT-GAN consisting of a novel graph translator and conditional graph discriminator for learning a conditional distribution of target graphs given the input graphs.

- We propose a novel graph encoder consisting of "edge convolution" layers that extract various relations among nodes containing both local and global information, and "node convolution" layers that embed the node representations based on the extracted relations.

- We propose a novel graph decoder consisting of the "edge deconvolution" and "node deconvolution" layers, which can decode the node representations first into the latent relations of the target graph and then generate the final target graph. The graph skip-connection is also utilized to map the learned latent relations between the input and target graphs.

- Extensive experiments have been conducted on both synthetic and real-world datasets on eight performance metrics to demonstrate the effectiveness and efficiency of the proposed model.

## 2 RELATED WORKS

**Graph Neural Networks**. The recent surge of research into GNN (Graph Neural Networks) can be generally divided into two categories: Graph Recurrent Networks and Graph Convolutional Networks. Graph Recurrent Networks originate from early work by Gori et al. (2005); Scarselli et al. (2009) and are based on recursive neural networks that have been extended by modern deep learning techniques such as gated recurrent units (Li et al., 2016). The other category, Graph Convolutional Networks, originate from spectral graph convolutional neural networks (Bruna et al., 2014), which were then extended by Defferrard et al. (2016) using fast localized convolutions, and further approximated by an efficient architecture for a semi-supervised setting proposed by Kipf & Welling (2017). Self-attention mechanism and subgraph-level information are also explored later to further improve the representation power of learned node embeddings (Veličković et al., 2017; 2018; Bai et al., 2019).

**Graph generation**. Most of the existing GNN based graph generation for general graphs have been proposed in the last two years and are based on VAE (Simonovsky & Komodakis, 2018; Samanta et al., 2018) and generative adversarial nets (GANs) (Bojchevski et al., 2018), among others (Li et al., 2018; You et al., 2018). Most of these approaches generate nodes and edges sequentially to form a whole graph, leading to the issues of being sensitive to the generation order and very time-consuming for large graphs. Differently, GraphRNN (You et al., 2018) builds an autoregressive generative model on these sequences with LSTM model and has demonstrated its good scalability.

**Data Translation involved Graphs**. A variety of graph-to-sequence models have been proposed to cope with different tasks including machine translation (Beck et al., 2018; Bastings et al., 2017), semantic parsing (Xu et al., 2018a;b; Song et al., 2018), and question generation (Chen et al., 2019), and health status prediction (Gao et al., 2019). The sequence-to-graph algorithms are generally popular with those working on NLP methods, including generating dependency graphs (Gildea et al., 2018; Wang et al., 2018) and AMR structures (Peng et al., 2018). A few of very recent attempts have also been made to develop graph-to-graph translation models. Jin et al. (2018b) proposed a domain-specific graph translation model to deal with molecular optimization task by utilizing the domain knowledge - junction tree and molecule graph. Do et al. (2019) dealt with the chemical reaction product prediction problem by predicting the reaction sequences based on the input graph of molecules. Sun & Li (2019) proposed a RNN based model for encoding and decoding the directed acyclic graph (converted from regular graphs), which can be viewed as a contemporary work to our work. However, this method is trained following the encoder-decoder architecture but in a supervised setting instead of learning a distribution of graphs. More importantly, it is difficult to scale to even modest-scale graph due to its one-node-per-step generation manner.

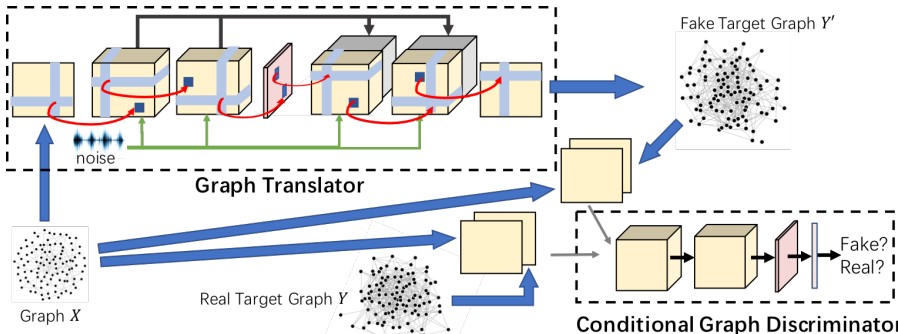

Figure 1: GT-GANs consisting of a graph translator and a conditional graph discriminator. A novel graph encoder and decoder are designed for the graph translation problem.

# 3 THE OVERALL ARCHITECTURE OF GT-GAN

In this section, we first present our problem formulation of graph translation problem. We then propose our new GT-GAN model for graph translation and discuss each component in detail in the subsequent sections.

## 3.1 PROBLEM FORMULATION FOR DEEP GRAPH TRANSLATION

Our goal is to learn an end-to-end translation mapping from an *input graph* to a *target graph*. Let an input graph $G_X = (\mathcal{V}, \mathcal{E}, A, S)$ such that $\mathcal{V}$ is the set of $N$ nodes, $\mathcal{E} \subseteq \mathcal{V} \times \mathcal{V}$ is the set of edges, and $A \in \mathbb{R}^{N \times N}$ is an adjacency matrix (binary or weighted), where $G_X$ can be weighted or unweighted, directed or undirected. Let $S \in \mathbb{R}^{N \times F}$ be a node feature matrix with each row representing a node feature vector $S_i$. Denote $e_{i,j} \in \mathcal{E}$ as an edge from the node $v_i \in \mathcal{V}$ to $v_j \in \mathcal{V}$; $A_{i,j} \in A$ therefore denotes the corresponding weight of the edge $e_{i,j}$. Similarly, we define a *target graph* $G_Y = (\mathcal{V}', \mathcal{E}', A', S')$ that shares the same node sets and node features with $G_X$ but with different topology and connection weights. Formally, *graph translation* is to learn a translator from an input graph $G_X \in \mathcal{G}_X$ with a random noise $U$ to generate a target graph $G_Y \in \mathcal{G}_Y$, where $\mathcal{G}_X$ and $\mathcal{G}_Y$ denote the domains of the input and target graphs, respectively. The translation mapping is denoted as $\mathcal{T}: U, G_X \rightarrow G_Y$.

Note that since our aim is to learn a conditional distribution of the target graphs given an input graph, we can cast the graph translation as a conditional graph generation problem, where an input graph can be mapped into any target graph that may have different topologies yet follow the same distribution. In contrast, the graph generation, that are designed to learn a distribution of graphs and generate a new graph sample based on this distribution, typically uses variational autoencoder framework for graph generation. Therefore, the previous graph generation frameworks such as graphVAE (Simonovsky & Komodakis, 2018) and GraphRNN (You et al., 2018) do not directly fit into "translation" setting.

**The Proposed GT-GAN Framework**. Fig.1 shows our proposed generic GAN framework for graph translation that consists of a graph translator $\mathcal{T}$ and a conditional graph discriminator $\mathcal{D}$. In this figure we assume the node feature has only one dimension for simplicity. Since our task is to train a conditional generator with "one-to-many mapping" instead of a deterministic one, the noise $U$ is introduced by the dropout function (Seltzer et al., 2013) in each convolution and deconvolution layer, as shown (in green lines) in Fig.1. Our graph translator $\mathcal{T}$ is trained to produce target graphs that cannot be distinguished from "real" ones by our conditional graph discriminator $\mathcal{D}$. Specifically, the generated target graph $G_{Y'} = \mathcal{T}(G_X, U)$ cannot be distinguished from the real one, $G_Y$, based on the current input graph $G_X$. $\mathcal{T}$ and $\mathcal{D}$ undergo an adversarial training process based on input and target graphs by solving the following the loss function:

$$\mathcal{L}(\mathcal{T}, \mathcal{D}) = \mathbb{E}_{G_X, G_Y}[\log \mathcal{D}(G_Y|G_X)] + \mathbb{E}_{G_X, U}[\log(1 - \mathcal{D}(\mathcal{T}(G_X, U)|G_X))], \quad (1)$$

where $\mathcal{T}$ tries to minimize this objective while an adversarial $\mathcal{D}$ tries to maximize it, i.e. $\mathcal{T}^* = \arg\min_{\mathcal{T}} \max_{\mathcal{D}} \mathcal{L}(\mathcal{T}, \mathcal{D})$. We also mix the GAN loss with the L1 loss to enforce sparsity similarity, which is also found useful in image translation problem (Isola et al., 2017),

$$\mathcal{L}_{l1}(\mathcal{T}) = \mathbb{E}_{A, A', U}[\|A' - T(G_X, U)\|_1], \quad (2)$$

where $T(G_X, U)$ refers to the adjacent matrix of generated graph. The training process is a trade-off between $\mathcal{L}_{l1}$ and $\mathcal{L}(\mathcal{T}, \mathcal{D})$, which jointly enforces $\mathcal{T}(G_X, U)$ and $G_Y$ to follow a similar, but not necessarily identical topological pattern. Specifically, $\mathcal{L}_{l1}$ makes $\mathcal{T}(G_X, U)$ share the same rough outline of sparsity pattern as $G_Y$, while $\mathcal{L}(\mathcal{T}, \mathcal{D})$ allows $\mathcal{T}(G_X, U)$ to vary to some degree. Thus, the optimal objective $\mathcal{T}^*$ of the translator, which generates graphs that are as "real" as possible, is defined as:

$$\mathcal{T}^* = \arg\min_{\mathcal{T}} \max_{\mathcal{D}} \mathcal{L}(\mathcal{T}, \mathcal{D}) + \mathcal{L}_{l1}(\mathcal{D}), \tag{3}$$

The graph translator $\mathcal{T}$ is an encoder-decoder architecture, where we propose a new graph encoder to obtain the node representations of the input graph and propose the graph deconvolution with skips to generate the target graph, as shown in Fig.1, which we elaborated in the followings sections.

## 3.2 GRAPH ENCODER

The graph encoder aims to learn the representations of nodes based on the node features and graph topology of the input graph. One of crucial challenges is to learn both local and global information in the graph embedding. For instance, when learning translation between two scale-free graphs, one needs to translate both the local information (i.e. n-hop neighborhood of each node) and the scale-free property (i.e. node degree distributions of whole graph) from an input graph to a target graph.

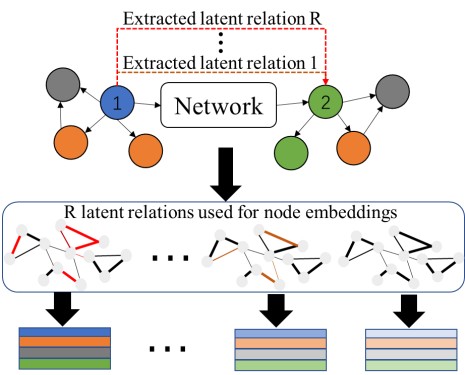

Figure 2: Latent relations for graph convolution

**The Proposed Graph Convolution**. To learn the local information, the proposed encoder learns each node representation based on its n-hop neighbors. To learn the global information, it learns each node representation by looking for more "virtual neighbors" regarding the latent relations from the aspect of the whole graph. As shown in Fig 2, though Node 1 and 2 are separated far away in the original network, they have the similarities in some properties, such as neighborhood structure and node degree. These nodes are treated as virtually connected ("virtual neighbors"). Thus, we first propose the "edge convolution" layers to learn a group of multi-mode relations from the topology of the input graph, which can include both the n-hop connections and the latent relations that are derived from their adjacent edges/relations. And then the "node convolution" layer is used to embed each node representations by aggregating its "virtual neighbors" that related to each latent relations. Fig.3 illustrates the details of these matrix operations involving graph convolution.

In each "edge convolution" layer, each node pair's latent relation is computed by its adjacent edges or the extracted adjacent relations from the last layer. In the directed graph, each node has incoming edge(s) and out-going edge(s). Thus, there are two learnable parametric vectors $\phi$ and $\psi$ as convolution filters for two directions to convolute the adjacent edges/relations for each node pairs. The relation $E_{i,j}^{l,m}$ in the $m$th relation mode of the $l$th layer is learned by the out-going edges/relations of node $v_i$ and the in-coming edges/relations of node $v_j$,

$$E_{i,j}^{l,m} = \sum_{n=1}^{R_{l-1}} (\sigma(\sum_{k_1=1}^{N} E_{i,k_1}^{l-1,n} \phi_{k_1}^{l,m}) + \sigma(\sum_{k_2=1}^{N} E_{k_2,j}^{l-1,n} \psi_{k_2}^{l,m})) \tag{4}$$

where $E_{i,j}^{1,1} \equiv A_{i,j}$ and $\phi^{l,m} \in \mathbb{R}^{N \times 1}$ refers to the filter vector to be learned and $\phi_{k_1}^{l,m}$ refers to the element of $\phi^{l,m}$ that is related to node $v_{k_1}$. $R_{l-1}$ refers to the number of relation modes extracted for the $(l-1)$th layer of the graph encoder.

After learning the various modes of relations, the "node convolution" layer learns each node's representations by aggregating its "virtual neighbors" in terms of each mode of relation. The $m$th feature vector of node representation tensor $\bar{H}_i^m \in \mathbb{R}^{1 \times F}$ for node $v_i$ is computed as:

$$\bar{H}_i^m = \sum_{n=1}^{R_{l-1}} (\sigma(\sum_{k_1=1}^{N} E_{i,k_1}^{l-1,n} \mu_{k_1}^m S_{k_1}) + \sigma(\sum_{k_2=1}^{N} E_{k_2,i}^{l-1,n} \nu_{k_2}^m S_{k_2})), \tag{5}$$

where $\bar{H}_i \in \mathbb{R}^{R_l \times F}$ and $R_l$ refers to the number of feature vectors in the "node convolution" layer. Here $\mu^m, \nu^m \in \mathbb{R}^{N \times 1}$ refer to the filter vectors for the two directions to be learned and $\mu_{k_1}^m$ refers to the element of $\mu^m$ that is related to node $v_{k_1}$. $\bar{H}_i$ is then flattened and transformed into a node representation vector $H_i \in \mathbb{R}^{1 \times C}$ by a fully connected layer. $C$ is the length of the node representation. Note that our graph encoder is designed for a directed graph, and it is easily generalized to an undirected graph, where the weight vector is shared by both directions.

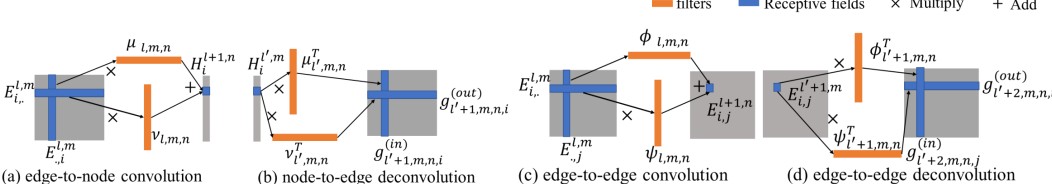

(a) edge-to-node convolution  (b) node-to-edge deconvolution  (c) edge-to-edge convolution  (d) edge-to-edge deconvolution

Figure 3: Matrix operations for graph convolution and graph deconvolution. In convolution operations, we need to utilize row filter to convolute "incoming" edges and column filter for "outgoing" edges. However, in deconvolution operations, we have to utilize the transposed filters, namely column filter to decode for "incoming" edges and row filter to decode for "outgoing" edges."

## 3.3 GRAPH DECODER

The decoder aims to generate the edges of the target graph by taking the extracted latent information of the input graph. It is straightforward to directly use the embedded node representation of the last layer to generate the target graph. However, the extracted information from each layer in the encoder could also be useful for generating the target graph. Thus, we consider all possible information learned in the encoder to be fed into a graph decoder.

Motivated by these observations, we propose a graph U-Net consisting of graph skips and dedicated graph deconvolution layers. The graph deconvolution decodes the single node (or edge) information to yield its incoming and outgoing adjacent edges as a mirrored graph convolution process. In addition, several skips are implemented to map the learned information of each layer in the encoder to mirror the corresponding layers in the decoder. Similar Graph U-Net was proposed in (Gao & Ji, 2019). The key difference is that their U-Net is barely a graph embedding method by using the old graph topology from pooling part to embed nodes during unpooling part. However, our Graph U-Net can not only do node embedding in graph encoder but also generate the new graph's topology in the graph decoder, which is necessary for the graph translation problem.

**The proposed Graph Deconvolution**. The proposed graph deconvolution technique incorporates both "node deconvolution" and "edge deconvolution" layers. First, the "node deconvolution" layer are used to generate the latent multi-mode relations of the target graph based on the learned latent node representations. As shown in Fig. 2(c), "node deconvolution" is a reversed process of the "node" convolution. Since each node has an influence to its relations connecting to other nodes. Then the relation $E_{i,j}^{1,m}$ between node $v_i$ and node $v_j$ in the $m$th relation mode of the $l$th "node" deconvolution layer in the decoder can be computed as follows:

$$E_{i,j}^{1,m} = \sum\nolimits_{n=1}^{C} \left( \sigma(H_i^n \bar{\mu}_j^m) + \sigma(H_j^n \bar{\nu}_i^m) \right), \quad (6)$$

where $\sigma(H_i^n \bar{\mu}_j^m)$ means the deconvolution contribution of node $v_i$ to its relation with node $v_j$ made by the $n$th element of its node representations, and $\bar{\mu}_j^m$ represents the element of the deconvolution filter vector $\bar{\mu}^m \in \mathbb{R}^{1 \times N}$ that is related to node $v_j$.

We can now recursively apply our proposed "edge deconvolution" layer to decode the latent relation between each pair of nodes from the upper layer to those of lower layer. As a reversed way of "edge" convolution, the relation of each pair of nodes in the $(l-1)$th layer can make contribution to generating itself and its adjacent relations in the $l$th layer, as shown in Fig. 2(d). Thus, the relation $E_{i,j}^{l,m}$ between node $v_i$ and node $v_j$ in the $l$th layer is computed as follows:

$$E_{i,j}^{l,m} = \sum\nolimits_{n=1}^{R'_{l-1}} \left( \sigma(\bar{\phi}_j^{l,m} \sum\nolimits_{k_1=1}^{N} E_{i,k_1}^{l-1,n}) + \sigma(\bar{\psi}_j^{l,m} \sum\nolimits_{k_2=1}^{N} E_{k_2,j}^{l-1,n}) \right), \quad (7)$$

where $\bar{\phi}^{l,m} \sum_{k_1=1}^{N} E_{i,k_1}^{l-1,n}$ is interpreted as the decoded contribution of node $i$ to its relations with node $v_j$, and $\bar{\phi}^{l,m}$ refers to the element of deconvolution filter vector that is related to node $v_j$. $R'_{l-1}$ refers to the number of relation modes extracted by the $(l-1)$th layer in the graph decoder. The output of the last "edge" deconvolution layer denotes the edges of the target graph.

**Skips for graph deconvolution**. Based on the graph deconvolution above, it is possible to utilize skips to link the extracted latent relation sets of each layers in the graph encoder with those in the graph decoder. Specifically, the output of the $l$th "edge deconvolution" layer with $R_l$ channels in the decoder is concatenated with the output of the $l$th "edge convolution" layer with $R'_l$ channels in encoder to form joint $R_l + R'_l$ channels, which are then input into the $(l+1)th$ deconvolution layer.

### 3.4 Conditional Graph Discriminator

The graph discriminator must distinguish between the "translated" target graph and the "real" ones based on the input graphs, as this helps to train the generator in an adversarial way. Technically, this requires the discriminator to accept two graphs simultaneously as inputs (a target graph and an input graph or a generated graph and an input graph), and classify the two graphs as either related or not. Thus, we propose a *conditional graph discriminator* (CGD) which leverages the same graph convolution layers in the translator for the graph classification, as shown in Fig.1. Specifically, the input and target graphs are both ingested by CGD and stacked into a $N \times N \times 2$ tensor which can be considered a 2-channel input. After obtaining the node representations, the graph-level embedding is computed by summing these node embeddings. Finally, a softmax layer is implemented to distinguish the input graph-pair from the real graph or generated graph.

### 3.5 Computational complexity analysis

The graph encoder and decoder shares the same time complexity. Without loss of generality, we assume all the hidden layers have the same number of feature maps as $M$. $P$ is the length of the fully connected layer in CGD. The worst-case total complexity of GT-GAN (i.e., the dense graph) is now $O(9N^2M^2 + 3N^2M^2 + N^2MP)$, where the first, second, and third terms represent "edge convolutions", "node convolutions", and fully connected layers in the graph discriminator, respectively. Similarly, the total memory consumption for GT-GAN is $O((9NM^2 + 9N^2M) + (3NM^2 + 3NM) + (N^2MP + P))$. In practice, many graphs are likely be sparse, thus it further reduces the computational and memory cost to $O(N)$ by using sparse matrix-vector operations (You et al., 2018), which paves the way toward modest scale graphs with hundreds or thousands of nodes, compared to most of existing graph generation methods, which often have $O(N^3)$ or even $O(N^4)$.

## 4 Experiment

This section reports the results of extensive experiments and ablation studies carried out to test the performance of GT-GAN on two synthetic and two real-world datasets. All experiments were conducted on a 64-bit machine with Nvidia GPU (GTX 1070,1683 MHz, 8 GB GDDR5). The code and data utilized are available at `https://github.com/anonymous1025/Deep-Graph-Translation-`.

### 4.1 Datasets

The experimental settings for each dataset were as follows. The rules for generating synthetic input-target graph pairs and the process of collecting the real-world graphs is provided in Appendix.

**Two synthetic datasets:** Two groups of synthetic datasets were used to validate the performance of the proposed GT-GAN: a scale-free graph dataset and a Poisson-random graph dataset. Each group has five subsets with different graph sizes (number of nodes): 10, 20, 50, 100 and 150. Each subset consists of 5000 input-target graph pairs; 2500 pairs were used for training and the remaining 2500 for testing.

**User authentication datasets**. The goal of this application was to forecast future potential malicious authentication graphs given the user's normal authentication graph. Each user authentication graph is a directed weighted graph, where nodes represent computers and the weights of the edges represent the authentication activities at certain frequencies. There are 78 pairs of graphs (malicious and normal behavior) of graph size 50 and 315 pairs of graphs of graph size 300 from 97 users in two subsets. We performed a 2-fold cross-validations and 3-fold cross-validation, respectively, for the two subsets.

Table 1: Evaluation results for the scale-free graphs

| Size | Methods | JS | HD | BD | WD | En-dist | C-dist | wl-sim | lt-sim |
|------|---------|-----|-----|-----|-----|---------|--------|--------|--------|
| 10 | Random-VAE | 0.42 | **0.98** | Inf | 7.58 | 0.3787 | 0.4528 | 0.3333 | 0.2494 |
| | GraphRNN | 0.47 | **0.98** | Inf | 1.64 | 0.7226 | 0.5319 | 0.2470 | 0.0055 |
| | GraphVAE | 0.67 | 1.00 | Inf | 2.85 | 0.6849 | 0.6664 | 0.3723 | 0.1576 |
| | GraphGMG | 0.43 | **0.98** | Inf | 1.69 | 0.6849 | 0.4763 | 0.3701 | 0.0120 |
| | S-Generator | **0.35** | 0.98 | 3.45 | 0.80 | 0.2097 | 0.2465 | 0.4185 | 0.5431 |
| | GT-GAN | **0.35** | 0.98 | **3.44** | **0.77** | **0.2034** | **0.2379** | **0.4195** | **0.5469** |
| 20 | RandomVAE | 0.51 | 0.97 | Inf | 1.74 | 0.4513 | 0.5400 | 0.3333 | 0.3813 |
| | GraphRNN | 0.50 | 0.98 | Inf | 1.44 | 0.7222 | 0.6087 | 0.2652 | 0.2373 |
| | S-Generator | 0.36 | **0.96** | 2.84 | 0.67 | **0.1367** | 0.1903 | 0.4665 | 0.7017 |
| | GT-GAN | **0.35** | **0.96** | **2.74** | **0.66** | **0.1367** | **0.1894** | **0.4681** | **0.7018** |
| 100 | GraphRNN | 0.48 | 0.88 | Inf | 0.90 | 0.7147 | 0.6519 | 0.2713 | 0.2138 |
| | S-Generator | **0.14** | 0.68 | 0.64 | **0.30** | **0.1149** | **0.1501** | 0.3522 | 0.8891 |
| | GT-GAN | 0.15 | **0.43** | **0.24** | 0.31 | 0.1153 | 0.2087 | **0.4078** | **0.9217** |
| 150 | GraphRNN | 0.42 | 0.74 | Inf | 0.95 | 0.7494 | 0.6266 | 0.2891 | 0.1874 |
| | S-Generator | 0.08 | 0.31 | **0.11** | 0.29 | 0.0949 | **0.1101** | 0.3493 | 0.8493 |
| | GT-GAN | **0.07** | **0.30** | **0.11** | **0.27** | **0.0931** | 0.2105 | 0.3926 | 0.8714 |

**Internet of Things (IOT) datasets** . This application focused IOT network malware confinement prediction (predicting optimal network operation given a compromised one). There are three subsets of graph pairs with different sizes (20, 40 and 60), where the nodes represent devices and the node attributes indicating the compromised status of the nodes. The weights of the edges represent the distance between two devices. There are 334 pairs of input (compromised IOT) and target graphs (optimal IOT) in each subset and each is divided into two parts for the 2-fold cross-validation.

## 4.2 BASELINE METHODS

We compare our GT-GAN against five state-of-the-art graph generation methods: 1) GraphRNN (You et al., 2018) is a new graph generation method based on sequential generation with the LSTM model; 2) GraphVAE (Simonovsky & Komodakis, 2018) is a probability-based graph generation method for small graphs; 3) GraphGMG (Li et al., 2018)is a framework based upon graph neural networks for small single graphs; 4) RandomVAE (Samanta et al., 2018) was described earlier; and 5) S-Generator is the part of our full model GT-GAN, which essentially is a graph translator with L1 loss but no discriminator. We propose this S-Generator model in order to evaluate the necessity of the proposed GT-GAN framework to learn the one-to-many mappings. All the comparison methods were trained on the malicious graphs without conditioning on the input graphs due to the models' inherent capability limitations. The datasets were assigned to each comparison model for the experiment based on their scalability in terms of graph size.

## 4.3 EVALUATION RESULTS ON SYNTHETIC DATASETS

**Results for the synthetic datasets**. To evaluate the similarity between the generated and real target graphs for scale-free dataset, we selected eight performance metrics: 1) two metrics are distances between generated and real graph in terms of Eigenvector centrality (En-dist) (Bonacich, 1987) and Closeness centrality (C-dist) (Freeman, 1978), where the lower the distance, the better the performance; 2) two metrics are similarity score based on the graph kernels of Weisfeiler Lehman kernel(wl-sim) (Shervashidze et al., 2011) and Lovasz Theta Kernel(lt-sim) (Johansson & et al, 2014), where the higher the score, the better the performance; 3) four metrics are used to evaluate the the node degree distribution correlation between the generated and real target graphs by: Jensen-Shannon distances (JS), the Hellinger Distance (HD), the Bhattacharyya Distance (BD) and the Wasserstein Distances (WD), where the lower the score, the better the performance.

As shown in table 1, our GT-GAN consistently outperforms all other baselines by a large margin, especially when the graph size becomes large (i.e.having the superiority of 34.6% than other methods when size is 150). The "Inf" entries represent distance over 1000. S-Generator is generally the second best methods in terms of these four evaluation metrics, highlighting the effectiveness of our proposed graph encoder and decoder.

To verify whether GT-GAN can indeed discover the underlying ground-truth translation rules between input-target pairs, we draw the node degree distribution curve for three pairs of generated and real target graphs by GT-GAN, as shown in Fig. 4. The curves of the generated graphs closely follow

the power-law rule and become even closer to the real graphs as the graph size increases, which is consistent with the findings in Table 1. This demonstrates that our GT-GAN model successfully learns the inherent properties of scale-free graphs during translation. Similar observations for the evaluation metrics (e.g. average degree, repository and density) of the Poisson random datasets and remaining scale-free subsets can be found in Appendixes B and C.

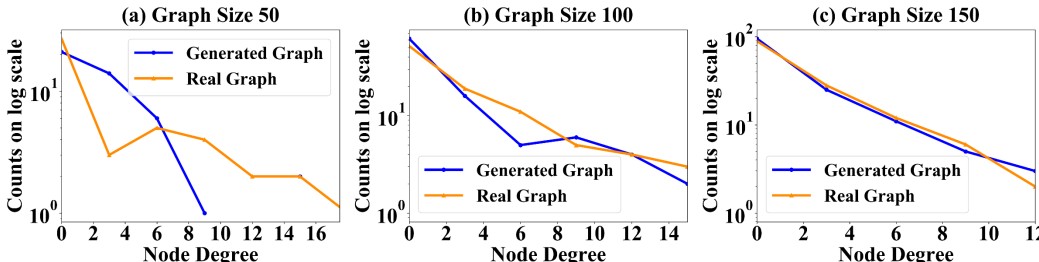

Figure 4: Examples of node degree distributions of generated and target graphs for scale-free graphs

## 4.4 EVALUATION RESULTS ON REAL APPLICATION DATASETS

**Results for the user authentication datasets**. For the real world dataset, we design an indirect evaluation metric inspired from a real-world classification problem: label imbalance issues. For example, we may want to build a classifier to determine whether an authentication graph of a user is malicious (positive) or normal (negative), but this user has few malicious records. For this difficult task, the graphs (i.e., malicious graphs) generated by GT-GAN, which has been trained on other users' records, can be utilized as positive samples to train the classifier. Specifically, when evaluating, the test set is further split evenly into two subsets. The first subset is used to train a graph classifier, as proposed by Nikolentzos et al. (2017), using only the normal graphs plus the generated malicious graphs. The second subset, which contains both the normal and real malicious graphs can then be used to validate the trained classifier. In addition, a "gold standard" classifier trained on both normal and *real* malicious graphs acts as the "best-possible-performer" and is used to evaluate all the different generative models to judge how "real" the graphs they generate are. We refer readers to the detailed evaluations in Appendix E.

Table 2: User authentication datasets

| Size | Method | P | R | AUC | F1 |
|---|---|---|---|---|---|
| | RandomVAE | 0.32 | 0.51 | 0.26 | 0.39 |
| | GraphRNN | 0.34 | 0.36 | 0.50 | 0.36 |
| 50 | S-Generator | 0.72 | 0.61 | 0.74 | 0.66 |
| | **GT-GAN** | **0.79** | **0.68** | **0.78** | **0.73** |
| | *Gold Standard* | *0.97* | *0.97* | *0.97* | *0.97* |
| | S-Generator | 0.77 | 0.58 | 0.62 | 0.66 |
| 300 | **GT-GAN** | **0.84** | **0.66** | **0.79** | **0.74** |
| | *Gold Standard* | *0.98* | *0.96* | *0.97* | *0.97* |

Table 3: IOT datasets

| Size | Method | R2 | MSE | P | ACC |
|---|---|---|---|---|---|
| | GraphRNN | 0.16 | 1775.58 | 0.23 | 83.97% |
| 20 | GraphVAE | 0.39 | 2109.64 | 0.32 | 81.19% |
| | GT-GAN | **0.67** | **370.91** | **0.85** | **92.00%** |
| | GraphRNN | 0.44 | 1950.46 | 0.29 | 70.54% |
| 40 | GraphVAE | **0.73** | 2410.57 | 0.16 | 66.60% |
| | GT-GAN | 0.69 | **408.50** | **0.86** | **93.94%** |
| | GraphRNN | 0.52 | 1831.43 | 0.04 | 61.07% |
| 60 | GraphVAE | 0.00 | 2453.61 | 0.04 | 50.64% |
| | GT-GAN | **0.62** | **566.88** | **0.80** | **94.63%** |

As shown in Table 2, classifiers trained by the graphs generated by GT-GAN can classify normal and hacked behaviors effectively with AUC above 0.78, which is well above the 0.5 obtained using a random model. GT-GAN significantly outperforms other methods by around 25%, 16%, 24.5% and 22.1%, respectively, on the four metrics: precision (P), recall (R), AUC and F1-score for the trained classifier. GT-GAN performs consistently better than other methods when the graph size rises from 50 to 300. In addition, GT-GAN clearly outperformed the S-Genertor in this evaluation setting. This confirms that using a translator alone to learn a deterministic output given an input graph is not sufficient to capture the generic distribution of the target graphs. In addition, the four direct evaluation mentioned above are also tested and the results can be found in Appendix C.

**Results on IOT dataset**. Table 3 compared the performance of GT-GAN and other comparison methods for the IOT dataset by examining the edges of the generated and real target graphs for four metrics: MSE (mean squared error), R2 (coefficient of determination score), Pearson Correlation (P) of adjacent matrix, and ACC (Accuray) for the correct existence of edges among all the pairs of nodes. The results show that GT-GAN performed almost the best for all the three subsets. GT-GAN got highest Pearson Correlation of around 0.8 for all three subsets compared to the other methods

which had Pearson Correlations below 0.4. Due to the L1-loss required to maintain the topology pattern similarity, GT-GAN also outperformed the comparison methods with around 8% ,26% and 40% superiority in ACC for the three subsets, respectively, and had the smallest MSE, at just one tenth of those achieved by comparison methods.

## 4.5 ABLATION STUDY ON THE GRAPH ENCODERS AND DECODER

Table 4: Ablation study on four datasets

| Dataset | Method | JS | HD | BD | WD | En-dist | C-dist | wl-sim |
|---|---|---|---|---|---|---|---|---|
| Scale-III | GCN+decoder | 0.18 | 0.48 | 0.27 | 18.84 | 0.6903 | 0.6751 | 0.4031 |
| | DCNN+decoder | 0.65 | 0.96 | Inf | 0.77 | 0.6907 | 0.6745 | 0.4032 |
| | Graph-U+decoder | 0.69 | 0.99 | Inf | 5.77 | 0.6931 | 0.6496 | 0.4040 |
| | Encoder+VGAE | 0.31 | 0.63 | 0.51 | 43.78 | **0.0922** | 0.2559 | 0.4003 |
| | GT-GAN | **0.15** | **0.43** | **0.24** | **0.31** | 0.1153 | **0.2087** | **0.4078** |
| | | P | R | AUC | F1 | En-dist | C-dist | wl-sim |
| Auth-I | GCN+decoder | 0.31 | 0.35 | 0.52 | 0.33 | 0.7394 | 0.7494 | 0.6632 |
| | DCNN+decoder | 0.59 | 0.55 | 0.55 | 0.57 | 0.0186 | 0.3349 | 0.6851 |
| | Graph-U+decoder | 0.41 | 0.60 | 0.30 | 0.49 | 0.6789 | 0.6859 | 0.9239 |
| | Encoder+VGAE | 0.49 | 0.46 | 0.61 | 0.47 | 0.0231 | 0.3129 | 0.6111 |
| | GT-GAN | **0.79** | **0.68** | **0.78** | **0.73** | **0.0134** | **0.1924** | **0.9439** |
| Auth-II | DCNN+decoder | 0.58 | 0.42 | 0.62 | 0.51 | **0.0007** | 0.1896 | 0.7033 |
| | Graph-U+decoder | 0.42 | 0.44 | 0.23 | 0.32 | 0.6931 | 0.6842 | 0.9744 |
| | GT-GAN | **0.84** | **0.66** | **0.79** | **0.74** | 0.0054 | **0.0681** | **0.9864** |
| | | R2 | MSE | P | ACC | En-dist | C-dist | wl-sim |
| IOT-III | GCN+decoder | 0.46 | 818.25 | 0.71 | 92.69 | 0.4990 | 0.4349 | 0.3304 |
| | DCNN+decoder | 0.52 | 721.98 | 0.74 | 93.26 | 0.3596 | 0.3217 | 0.3292 |
| | Graph-U+decoder | 0.45 | 826.63 | 0.70 | 92.46 | 0.3526 | **0.2771** | 0.3310 |
| | Encoder+VGAE | 0.12 | 1337.16 | 0.44 | 88.14 | 0.4811 | 0.4876 | 0.3333 |
| | GT-GAN | **0.62** | **566.88** | **0.80** | **94.63** | **0.3350** | 0.3051 | **0.3899** |

To further validate the superiority of the proposed graph convolution and deconvolution layers, an ablation experiment was conducted by replacing the encoder and decoder with node embedding and decoder methods normaly used. The graph encoder was replaced by the GCN (Kipf & Welling, 2017), DCNN (Atwood & Towsley, 2016) and Graph U-NET (Gao & Ji, 2019), both of which consider edge and node features for graph embedding. The graph decoder was replaced by the decoder in VGAE (Kipf & Welling, 2016). There were thus three method combinations for comparison.

Table. 4 shows the results of the ablation study of the proposed encoder and decoder on part of the scale-free (Scale), user authentication (Auth) and IOT datasets. There are two major findings here. First, the encoder of GT-GAN outperformed both the GCN- and DCNN- based encoders by a large margin on these datasets, especially for the real-world datasets, where the edges of the graphs can have a very complex meaning. For example, on Auth-I, GT-GAN performed 43%, 50%, 31%, and 38% better on average, when compared with the GCN and DCNN encoders in terms of precision, recall, AUC and F1-scores, respectively. Second, the proposed decoder in GT-GAN was deemed both effective and irreplaceable for graph generation. For example, on IOT-III, GT-GAN performed 6.97%, 45.00%, and 83.33% better than the decoder in VGAE in terms of ACC, P and R2, respectively, as well as a low MSE below 1000.

## 4.6 MODEL SCALABILITY ANALYSIS

We compare the scalability of GT-GAN against three graph generation methods as shown in Fig.5. Our GT-GAN model significantly outperforms other state-of-the-art baselines in terms of both computational time and memory consumption. As the graph size increases up to 50, both computational time and the memory consumption of the GT-GAN remains almost constant. In contrast, the runtime and memory consumption of RandomVAE and the runtime of GraphVAE increase super-linearly as the graph size increases, making it hard to scale even to a graph size of 50. Interestingly, the runtime and memory consumption of GraphRNN also increases only slightly as the graph size increases. However, our GT-GAN model achieves around ten times speedups while requiring almost half of memory, compared to GraphRNN, highlighting the strong linear complexity of GT-GAN in practice.

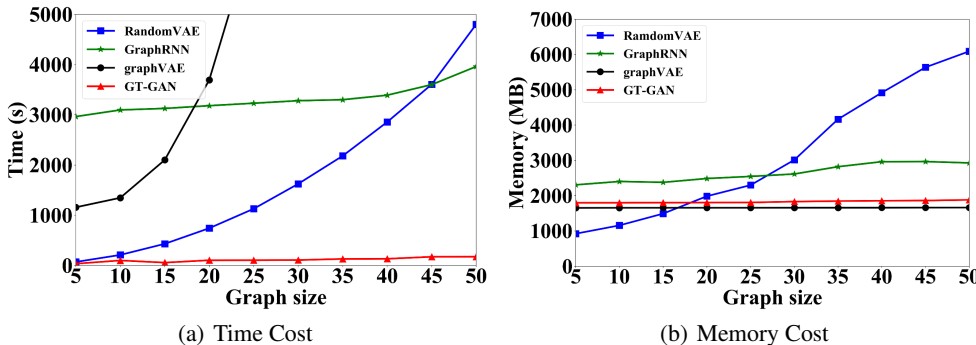

Figure 5: Scalability plots for memory and time cost of GT-GAN, RandomVAE, GraphVAE and GraphRNN

## 5 CONCLUSIONS

This paper focuses on a new problem: deep graph translation. To achieve this, we propose a novel GT-GAN which translates an input graph to a target graph. To learn both global and local mapping between graphs, a new graph encoder-decoder model have been proposed while preserving the graph patterns in various scales. Extensive experiments have been conducted on the synthetic and real-world dataset to compare with the state-of-the-art graph generation models. Experimental results show that our GT-GAN can discover the ground-truth translation rules, and significantly outperform other baselines in terms of both effectiveness and scalability. This paper opens a thread of research for deep graph translation in many practical applications.

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

## A MORE DESCRIPTIONS AND EXPERIMENTS FOR SCALE FREE DATASET

**Generation of graph pairs**. Each input graph is generated as a directed scale-free network, whose degree distribution follows power-law property Bollobás et al. (2003). A node will be selected as target node with probability proportional to its in-degree, which will be linked to a new source node with probability of 0.41. Similarly, a node will be selected as source node with probability proportional to its out-degree, which will be linked to a new target node with probability of 0.54. Then, in the target graph, each weight between two connected nodes will be added by $m$, where $m$ could be any value larger than 1. Thus, both input and target graphs are scale-free graphs.

**Indirect evaluation and ablation study**. Other than directly measuring the node degree distribution similarity between the generated and real target graphs, we also conduct an indirect evaluation as done for user authentication dataset. Table 5 shows the average results of graph classifiers: Precision, Recall, AUC, and F1-measure for different methods. For small graphs (e.g., nodes less then 10), the power-law property of scale-free networks is less obvious compared to the larger size graphs, which may explain why the tasks on smaller scale-free graphs are more difficult. However, when the sizes of graph increase, GT-GAN become more close to the performance of "Gold Standard" with average difference of 10%, 4%, 5% and 9% on F1 accordingly, and significantly outperforms other methods by large margin up to 51%, 35%, 10%, and 19% on F1, respectively.

Table 6 shows the ablation study on dataset of Scale-III and Scale-IV in both metric evaluation and indirect distribution evaluation.

Table 5: Indirect evaluation for scale-free graphs

| Dataset | Method | P | R | AUC | F1 |
|---|---|---|---|---|---|
| | RandomVAE | 0.83 | 0.29 | 0.31 | 0.42 |
| | GraphRNN | 0.31 | 0.11 | 0.49 | 0.16 |
| | GraphVAE | 0.75 | 0.23 | **0.65** | 0.35 |
| Scale-I | GraphGMG | 0.42 | 0.12 | 0.49 | 0.18 |
| | S-Generator | 0.46 | **0.83** | 0.43 | 0.59 |
| | GT-GAN | **1.00** | 0.50 | 0.52 | **0.67** |
| | Gold Standard | 0.81 | 0.74 | 0.82 | 0.77 |
| | RandomVAE | 0.50 | 1.00 | **0.54** | 0.66 |
| | GraphRNN | 0.67 | 0.12 | 0.50 | 0.21 |
| Scale-II | S-Generator | 0.50 | **1.00** | 0.50 | 0.67 |
| | GT-GAN | **1.00** | 0.50 | 0.50 | **0.67** |
| | Gold Standard | 0.76 | 0.67 | 0.72 | 0.71 |
| | RandomVAE | 0.89 | 0.67 | 0.84 | 0.76 |
| | GraphRNN | 0.52 | 0.53 | 0.70 | 0.52 |
| Scale-III | S-Generator | 0.50 | **1.00** | 0.37 | 0.67 |
| | GT-GAN | **0.93** | 0.82 | **0.94** | **0.87** |
| | Gold Standard | 0.94 | 0.90 | 0.97 | 0.91 |
| | GraphRNN | 0.61 | 0.65 | 0.67 | 0.60 |
| | S-Generator | 0.50 | **1.00** | 0.50 | 0.67 |
| Scale-IV | GT-GAN | **0.72** | 0.69 | **0.68** | **0.70** |
| | Gold Standard | 0.99 | 0.61 | 0.81 | 0.75 |
| | GraphRNN | 0.73 | **0.92** | 0.92 | 0.81 |
| | S-Generator | **1.00** | 0.50 | 0.50 | 0.67 |
| Scale-V | GT-GAN | 0.94 | 0.79 | **0.96** | **0.86** |
| | Gold Standard | 0.99 | 0.93 | 0.96 | 0.95 |

Table 6: Ablation study on Scale-free datasets

| Dataset | Method | Indirect | | | | Direct | | | |
|---|---|---|---|---|---|---|---|---|---|
| | | P | R | AUC | F1 | JS | HD | BD | WD |
| | GCN+decoder | 0.85 | 0.16 | 0.89 | 0.27 | 0.32 | 0.65 | 0.58 | 2.87 |
| Scale-III | DCNN+decoder | 0.64 | 0.93 | 0.83 | 0.76 | 0.68 | 0.99 | Inf | 5.59 |
| | Encoder+VGAE | 0.66 | 0.26 | 0.59 | 0.37 | 0.28 | 0.59 | 0.45 | 49.15 |
| | GT-GAN | 0.93 | 0.82 | 0.94 | 0.87 | 0.43 | 0.89 | 1.66 | 2.43 |
| | GCN+decoder | 0.74 | 0.51 | 0.80 | 0.60 | 0.18 | 0.48 | 0.27 | 18.84 |
| Scale-IV | DCNN+decoder | 0.50 | 1.00 | 0.83 | 0.67 | 0.65 | 0.96 | Inf | 0.77 |
| | Encoder+VGAE | 0.50 | 0.62 | 0.50 | 0.55 | 0.31 | 0.63 | 0.51 | 43.78 |
| | GT-GAN | 0.72 | 0.69 | 0.68 | 0.70 | 0.15 | 0.43 | 0.24 | 0.31 |

Fig. 10 shows 18 examples of the node degree distribution curve in generated and real target graphs for scale free dataset from size 50 to 150.

## B DESCRIPTIONS AND EXPERIMENTS FOR POISSON RANDOM DATASET

**Generation for graph pairs**. Each input graph is generated by Krapivsky & Redner (2001), which is a directed growing random network. Then for an input graph with $E$ number of edges, we randomly add k$E$ number of edges on it to form the target graph, where k follows the Poisson distribution with the mean of 5.

**Experiment results**. For Poisson random graphs, the distributions of $k$ in the real target graphs and those generated graphs are compared. The mean of edge increasing ratio $k$ for generated graphs by our GT-GAN is 3.6, compared to the real value of 5, which implies that the GT-GAN generally is able to discover the underlying

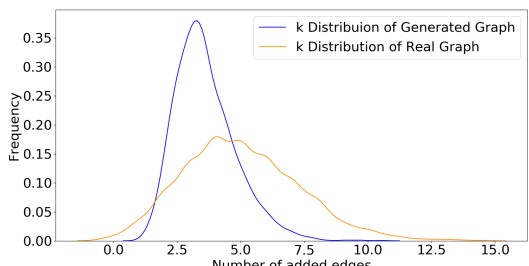

Figure 6: Distribution of $k$ for generated and real graphs in Poisson random dataset

increasing ratio between input and target graphs. More evaluation results (e.g. degree and repository) can be found in Appendix B. We draw the probability density curve of the proportion k. Fig. 6 shows the distribution of the $k$ in graphs generated by GT-GAN and the real graphs. The distribution plot is drew based on 3000 samples. Both of the two distribution have main degree values in the range from 2 to 7, while there is difference in the max frequency due to the limit of the samples amount. However, it prove that the proposed GT-GAN do learn the distribution type of translation parameter $k$ in this task.

Results for indirect evaluation on Poisson Random graphs are listed in Table 7. Though the Poisson random task is easier than scale-free graphs, the GT-GAN still outperforms others on AUC and F1, and its performance is highly close to the "Gold Standard". Table 8 shows the distance measurement between generated graphs and real graphs in several metrics. For the metric "degree", we use Wasserstein distances to measure the distance of two degree distribution. For other metrics, we calculate the MSE between generated graphs and real graphs.

Table 7: Distribution evaluation for Poisson random datasets

| Dataset | Method | P | R | AUC | F1 |
|---|---|---|---|---|---|
| Poisson-I | RandomVAE | 0.98 | 0.75 | **0.99** | 0.85 |
| | GraphRNN | 0.98 | 0.99 | **0.99** | **0.98** |
| | GraphVAE | 0.98 | 0.92 | 0.97 | 0.94 |
| | GraphGMG | 0.98 | 0.98 | 0.98 | **0.98** |
| | S-Generator | 0.50 | **1.00** | 0.50 | 0.66 |
| | GT-GAN | **1.00** | 0.87 | 0.94 | 0.90 |
| | Gold Standard | 0.99 | 1.00 | 1.00 | 0.99 |
| Poisson-II | RandomVAE | **1.00** | 0.70 | 0.99 | 0.82 |
| | GraphRNN | **1.00** | **1.00** | **1.00** | **1.00** |
| | S-Generator | **1.00** | **1.00** | **1.00** | **1.00** |
| | GT-GAN | **1.00** | 0.99 | **1.00** | 0.99 |
| | Gold Standard | 0.99 | 1.00 | 1.00 | 0.99 |
| Poisson-III | RandomVAE | 0.93 | 0.46 | **1.00** | 0.63 |
| | GraphRNN | **1.00** | 0.99 | 0.99 | 0.99 |
| | S-Generator | 0.49 | 0.98 | 0.35 | 0.65 |
| | GT-GAN | **1.00** | 0.99 | **1.00** | 0.99 |
| | Gold Standard | 0.99 | 1.00 | 1.00 | 0.99 |
| Poisson-IV | GraphRNN | **1.00** | 0.99 | **1.00** | 0.99 |
| | S-Generator | 0.50 | **1.00** | 0.51 | 0.66 |
| | GT-GAN | 0.90 | **1.00** | **1.00** | 0.94 |
| | optimal | 1.00 | 1.00 | 1.00 | 1.00 |
| Poisson-V | GraphRNN | 0.95 | 0.99 | **1.00** | 0.96 |
| | S-Generator | 0.50 | **1.00** | 0.49 | 0.66 |
| | GT-GAN | **0.97** | **1.00** | **1.00** | **0.98** |
| | Gold Standard | 1.00 | 0.99 | 1.00 | 0.99 |

Table 8: MSE of Graph properties measurements for Poisson random dataset

| Dataset | Method | Density | Ave-Degree | Reciprocity |
|---|---|---|---|---|
| Poisson-I | RandomVAE | 0.1772 | 2.8172 | 0.3917 |
| | GraphRNN | 0.2665 | 2.2078 | 0.1344 |
| | GrapgGMG | 0.3519 | 2.4286 | 0.1338 |
| | GraphVAE | 0.2881 | 3.1986 | 0.3103 |
| | S-Generator | **0.2993** | **1.5751** | **0.0737** |
| | GT-GAN | 0.3084 | 1.7707 | 0.1327 |
| Poisson-II | RandomVAE | 0.2078 | 7.0860 | 0.4182 |
| | GraphRNN | 0.2305 | 4.9256 | 0.1190 |
| | S-Generator | 0.2111 | 3.2207 | 0.0430 |
| | GT-GAN | **0.2013** | **3.2047** | **0.0388** |
| Poisson-III | RandomVAE | Inf | 23.680 | 0.5362 |
| | GraphRNN | **0.0110** | 3.6000 | 0.0125 |
| | S-Generator | 0.0120 | **2.9082** | 0.0125 |
| | GT-GAN | 0.0155 | 3.2960 | **0.0047** |
| Poisson-IV | GraphRNN | 0.0123 | 3.5475 | 0.0034 |
| | S-Generator | **0.0029** | **2.9167** | **0.0034** |
| | GT-GAN | 0.0142 | 4.3730 | 0.0043 |
| Poisson-V | GraphRNN | **0.0012** | 3.6619 | 0.0016 |
| | S-Generator | 0.0013 | **2.9467** | **0.0016** |
| | GT-GAN | 0.0061 | 5.0410 | 0.0019 |

## C  DESCRIPTION OF IOT DATASET

There are three sets of IoT nodes at different amount (20, 40 and 60) encompassing temperature sensors connected with Intel ATLASEDGE Board and Beagle Boards (BeagleBone Blue), communicating via Bluetooth protocol. Benign and malware activities are executed on these devices to generate the initial attacked networks as the input graphs. Benign activities include MiBench and SPEC2006, Linux system programs, and word processor. The real target graphs are generated by the classical malware confinement methods: stochastic controlling with malware detection.

## D  MORE EXPERIMENTAL RESULTS FOR USER AUTHENTICATION GRAPH SET

**About Original Dataset**  This dataset includes the authentication activities of 97 users on their accessible computers and servers in an enterprise computer network (Kent, 2015). Each user account generates a log file recording the computer accessing history, which could be formulated as a directed weighted graph called authentication graph, where nodes represent computers and the directed edges weights represent the authentication activities with certain frequencies. This data set spans one calendar year of contiguous activity spanning 2012 and 2013. It originated from 33.9 billion raw event logs (1.4 terabytes compressed) collected across the LANL enterprise network of approximately 24,000 computers. Here we consider two sub dataset. First is the user log-on activity set. This data represents authentication events collected from individual Windows-based desktop computers, servers, and Active Directory servers. Another dataset presents specific events taken from the authentication data that present known red team compromise events, as we call malicious event. The red team data can used as ground truth of bad behavior which is different from normal user. Each graph can represent the log-on activity of one user in a time window. The event graphs are defined like this: The

node refers to the computers that are available to a user and the edge represents the log-on activity from one computer to another computer of the user.

**Direct evaluation of User authentication Graph Set**. For the user authentication graphs, the real target graphs and those generated are compared under well-recognized graph metrics including degree of nodes, reciprocity, and density. We calculate the distance of degree distribution and Mean Sqaured Error (MSE) for reciprocity and density. Table 9 shows the mean square error of the generated graphs and real graphs for all users evaluated for both graph generation methods and ablation models.

Table 9: MSE of Graph properties measurements for user authentication dataset

| Dataset | Method | Density | Reciprocity | Ave-Degree |
|---|---|---|---|---|
| | RandomVAE | 0.0005 | 0.0000 | 6.4064 |
| | GraphRNN | 0.0032 | 0.0000 | 2.7751 |
| | S-Generator | 0.0244 | 0.0342 | 24.130 |
| Auth-I | GCN+decoder | 0.0006 | 0.0000 | 0.3510 |
| | DCNN+decoder | 0.0000 | 0.0000 | 0.0137 |
| | Encoder+VGAE | 1.1050 | 0.0000 | 0.1052 |
| | GT-GAN | **0.0003** | **0.0000** | **0.0002** |
| | S-Generator | 0.0113 | 0.0010 | 8.6839 |
| Auth-II | DCNN+decoder | 0.0000 | 0.0000 | 0.0039 |
| | GT-GAN | **0.0004** | **0.0000** | **0.0006** |

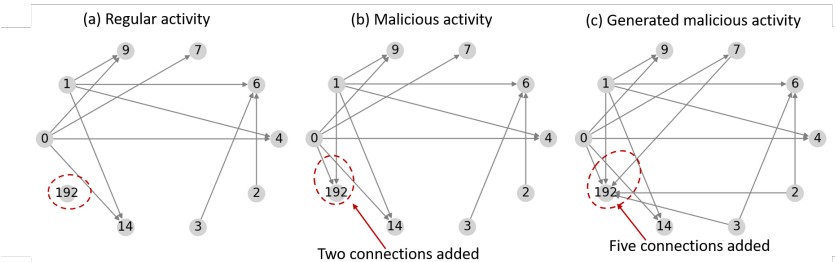

Figure 7: Regular graphs, malicious graphs and generated graphs of User 049

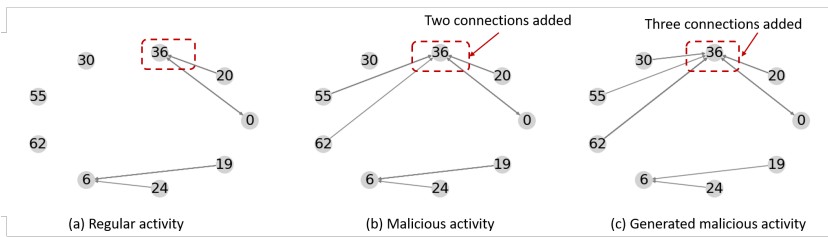

Figure 8: Regular graphs, malicious graphs and generated graphs for User 006

**Case Studies on the generated target graphs**.
Fig.7 shows the example of User 049 with regular activity graph, real malicious activity graph and malicious activity graph generated by our GT-GAN from left to right. Only those of edges with difference among them are drawn for legibility. It can be seen that, the hacker performed attacks on Computer 192, which has been successfully simulated by our GT-GAN. In addition, GT-GAN also correctly identified that the Computer 192 is the end node (i.e., with only incoming edges) in this attack. This is because GT-GAN can learn both the global hacking patterns (i.e., graph density, modularity) but also can learn local properties for specific nodes (i.e., computers). GT-GAN even successfully predicted that the hacker connect from Computers 0 and 1, with Computers 7 and 14 as false alarms. For User006 in Fig. 8, the red team attackers make more connections on Node 36 compared to user's regular activity, as marked in red rectangle. GT-GAN leans how to choose the Node 36 and it generated more connections too in the Node 36.

## E APPENDIX E: FLOWCHART OF INDIRECT EVALUATION PROCESS

Fig. 9 shows the process of the indirect evaluation process for evaluating whether the generated and the real target graphs follow the same distribution.

## F ARCHITECTURE PARAMETER FOR GT-GAN MODEL

**Graph Translator**: Given the graph size (number of nodes) $N$ of a graph. The output feature map size of

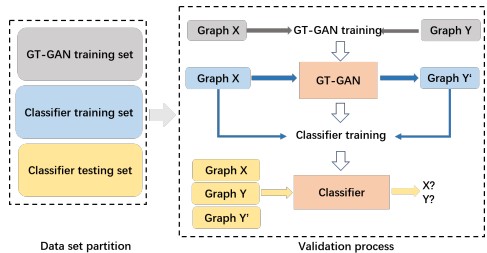

Figure 9: Flow chart of validation

each layer through graph generator can be expressed
as:
$N \times N \times 1 \rightarrow N \times N \times 5 \rightarrow N \times N \times 10 \rightarrow N \times 1 \times 10 \rightarrow N \times N \times 10 \rightarrow N \times N \times 5 \rightarrow N \times N \times 1$
**Discriminator**: Given the graph size (number of nodes)
$N$ of a graph. The output feature map size of each layer
through graph discriminator can be expressed as:
$N \times N \times 1 \rightarrow N \times N \times 5 \rightarrow N \times N \times 10 \rightarrow N \times 1 \times 10 \rightarrow 1 \times 1 \times 10$
For the edge to edge layers, the size of two kernels in two directions are $N \times 1$ and $1 \times N$. For the node to edge
layer, the kernel size is $1 \times N$

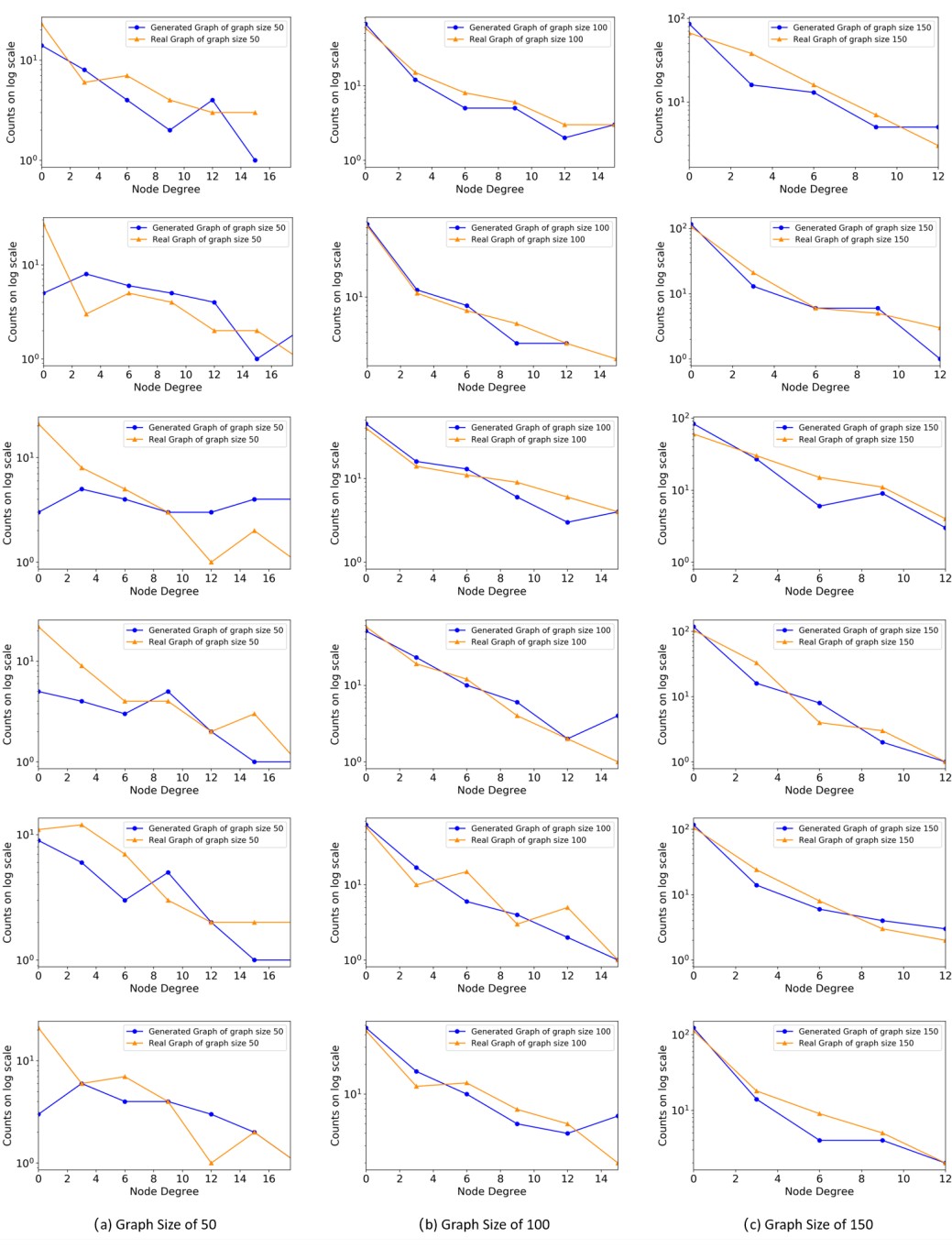

Figure 10: Examples of node degree distrbution for generated graphs and real graphs

