# OpenReview forum: "Deep Graph Translation"
_ICLR.cc/2020/Conference — Reject_

### Official Review · AnonReviewer3 · 2019-10-23
**Official Blind Review #3**

**Rating:** 3

**Review:**

This paper studies a new problem, i.e., generating graphs conditioned on an input graph. The authors proposed a new framework GT-GAN, which is composed of (i) a graph encoder to representation of the input graph; (ii) a generator to generate graphs; (iii) a discriminator to fool the discriminator so that the generated graph can be more realistic; and (iv) a l1-norm regularizer to make the generated graph similar to the target graph. Experimental results on synthetic and real-world datasets demonstrate the effectiveness of the proposed method for graph translation. Overall, the problem studied in this paper is interesting and novel, and the proposed method makes sense. There are some concerns about the paper and I would like to increase my rating if the authors can address my concerns:

(i) Could you double check if E_{ij}^{1,1}=A? The dimension seems to be problematic in Eq.(5) if E_{ij}^{1,1} is of dimension N x N. If E_{ij}^{1,1} is N x N, then E_{I,k1}^{l-1,n} is also N x N. In Eq.(5), mu_{k1}^m is 1 x 1 and S_{k1} is 1 x N. Please double check. E_{ij}^{1,1} is of dimension N x N

(ii) The explanation of why the proposed graph convolution can learn global information in the graph embedding is unclear. For example, how the embedding can preserve the scale-free property? Could you provide more explanations?

(iii) Though the studied problem is interesting, the proposed method makes sense but is not very novel. It seems to be adopting GAN with GNN and l1 regularizer.

(iv) The contribution of the l1 regularizer is not analyzed. What the performance will be if we remove the l1 regularizer?


**Experience Assessment:**

I have published one or two papers in this area.

**Review Assessment: Checking Correctness Of Derivations And Theory:**

I carefully checked the derivations and theory.

**Review Assessment: Checking Correctness Of Experiments:**

I assessed the sensibility of the experiments.

**Review Assessment: Thoroughness In Paper Reading:**

I read the paper thoroughly.

---

> ### Author Response · Authors · 2019-11-12
> **Authors' Response to Reviewer #3 (Part I)**
>
>
> Dear Reviewer,
>
> Thank you very much for the careful reading and valuable comments. Below we address the concerns mentioned in the review:
>
> --------------------------------------------
> Q1: Could you double check if E_{ij}^{1,1}=A? The dimension seems to be problematic in Eq.(5) if E_{ij}^{1,1} is of dimension N x N. If E_{ij}^{1,1} is N x N, then E_{I,k1}^{l-1,n} is also N x N. In Eq.(5), mu_{k1}^m is 1 x 1 and S_{k1} is 1 x N. Please double check. E_{ij}^{1,1} is of dimension N x N.
>
> Answer: Thank you very much for pointing it out. You are absolutely right and this is a typo. It should be E_{ij}^{1,1}=A_{i,j}. The E_{ij}^{1,1} is a scalar value that refers to one element in the adjacency matrix of the input graph.
>
> --------------------------------------------
> Q2: The explanation of why the proposed graph convolution can learn global information in the graph embedding is unclear. For example, how the embedding can preserve the scale-free property? Could you provide more explanations?
>
> Answer: The proposed graph convolution can learn global information because when performing each node embedding by edge-to-node layer, we rely on not only each node’s n-hop neighborhood but also its extracted latent neighborhoods (“Virtual neighbors”) from the whole graph, which could be far away from this node.
>
> For instance, assume we have two nodes 1 and 2. Though nodes 1 and 2 are not connected in the input graph, if they have the same node degree, then based on the latent relation of “node degree similarity”, they can be treated to be virtually connected. And node degree similarity is a very important criterion in scale-free graph since it indicates the knowledge that these two nodes have a similar probability to connect to others and that knowledge may be critical in generating the output graph. Thus, we allow our model to extract various latent relationships among nodes before performing node embedding aggregation. And these latent relations can also be directly mapped to the decoder by skip-connections. We have also added an example in Figure. 2 for illustration in the revision.

---

> ### Author Response · Authors · 2019-11-12
> **Authors' Response to Reviewer #3 (Part II)**
>
> --------------------------------------------
> Q3: Though the studied problem is interesting, the proposed method makes sense but is not very novel. It seems to be adopting GAN with GNN and l1 regularizer.
>
> Answer: Although it seems like it is “easy” to adopting GAN with GNN, we would like to argue that it is definitely “non-trivial” to combine these two for graph translation. In particular, in order to apply GAN on graph domains, we have designed a very novel conditional GAN framework for graph translation, namely GT-GAN which consists of a novel graph translator and conditional graph discriminator, both of which have been proposed for the first time (to the best of our knowledge).
>
> For the graph decoder, this is actually the most challenging part. We are the first to propose the graph deconvolution layer, which has not been studied and handled before. The existing graph decoders (VGAE [1], graphite [2]) are all about just calculating the similarity of node embedding to generate the edges. They are over-restrictive by assuming the similar node embeddings indicate the edges between them. Graph U-net [3] is designed just for graph embedding on a fixed graph topology without generating graph edges, which is different from ours.  As shown in Table 4, we performed the ablation study by replacing our graph decoder with other graph generation methods like VGAE and showed that our graph decoder consistently performed better.
>
> For graph encoder, our graph encoder is different from most of the graph encoders such as GCN, GGSNN, MPNN, GAT. These graph encoders shared the similar high-level methodology and the main differences are how to design the aggregation functions and how to pass the “messages” through a set of nodes or both a set of edges and nodes. Our graph encoder is quite different from the existing GNN variants. We first propose the “edge convolution” layers to learn a group of multi-mode relations from the topology of the input graph, which can include both the n-hop connections and the latent relations that are derived from their adjacent edges/relations. And then the “node convolution” layer is used to embed each node representations by aggregating its “virtual neighbors” that related to each latent relations extracted from different the adjacent edges/relations. As shown in Table 4, we performed the ablation study by replacing our graph encoder with others including GCN, DCNN, and Graph UNets and showed that our graph encoder consistently performed the best.
>
> Also, our conditional graph discriminator (CGD) was also proposed for the first time.  CGD is designed to accept two graphs simultaneously as inputs (a target graph and an input graph or a generated graph and an input graph), and classify the two graphs as either related or not. Furthermore, CGD leverages the same graph convolution layers in the translator for the graph classification.
>
> In summary, our decoder is the first graph deconvolution method and our encoder is different from existing methods. Our conditional graph discriminator is also novel. Our ablation studies (in Sec. 4.5)  have validated the superiority of our proposed graph deconvolution and encoder over the existing GNN-based encoders and decoders.
>
> [1] Kipf, T. N., & Welling, M. (2016). Variational graph auto-encoders. arXiv preprint arXiv:1611.07308.
> [2] Grover, A., Zweig, A., & Ermon, S. (2018). Graphite: Iterative generative modeling of graphs. arXiv preprint arXiv:1803.10459.
>
> --------------------------------------------
> Q4: The contribution of the l1 regularizer is not analyzed. What the performance will be if we remove the l1 regularizer?
>
> Answer: The contribution of L1 regularization is significant but does not dominate.
>
> We use the L1 regularization to enforce a rough outline of sparsity pattern and edge weight scale similarity between generated and real graphs, which is also found to be useful in image translation problems [3]. The training process is a trade-off between L1 loss and the discriminator loss, which jointly enforces the generated and real graphs to follow a similar, but not necessarily identical topological pattern. We described them in the last paragraph of Page 3 and the first paragraph of Page 4 in the paper.
>
> On one hand, we have tested the performance without l1 regularization, and the performance gets much worse. Thus we did not show them in the paper. But it highlights that the L1 is very important to the performance.
> On the other hand, as shown in Tables 1 and 2, our full model (GT-GAN) clearly outperforms S-Generator, which is essentially a graph translator with only L1 loss and without conditional graph discriminator. It clearly indicated that using L1 regularization alone is not enough to capture the generic distribution of the target graphs.

---

### Official Review · AnonReviewer1 · 2019-10-24
**Official Blind Review #1**

**Rating:** 3

**Review:**

This paper studies a problem of graph translation, which aims at learning a graph translator to translate an input graph to a target graph. The authors propose an adversarial training framework to learn the graph translator, where a discriminator is trained to discriminate between the true target graph and the translated graph, and the translator is optimized by fooling the discriminator. The authors conduct experiments on both synthetic and real-world datasets. The results prove the effectiveness and the efficiency of the proposed approach over many baselines.

Strengths:

1. The problem is new and well-motivated.
Data translation is an important problem and has been widely studied in many research domains such as computer vision and natural language processing. Despite the importance, the problem has not been thoroughly explored in the graph domain, and most existing studies only focus on standard graph generation problems. In this sense, this paper studies a very new problem, which is quite novel. Moreover, the problem is important, which can have many potential downstream applications on graph data. Overall, the problem is new and well-motivated.

2. The proposed approach is quite intuitive.
The paper proposes an adversarial training approach to the problem, where a graph translator is learned based on a graph discriminator. During training, the graph discriminator aims at discriminating between the true target graph and the translated graph conditioned on an input graph, and the graph translator is trained by fooling the discriminator. The graph translator is built on top of an encoder-decoder framework, where the encoder and decoder are parameterized by graph neural networks. Overall, the proposed method is quite reasonable, which is easy to follow.

3. The results are promising.
The authors conduct extensive experiments on both synthetic and real-world datasets, and compare the proposed approach against many strong baseline methods for graph generation. The results are quite promising, which prove both the effectiveness and the efficiency of the approach.

Weaknesses:

1. The novelty of the proposed approach is limited.
The proposed approach is mainly built on top of the adversarial training framework, where a graph neural network is used to parameterize the graph translator. For adversarial training, although it is very intuitive, such a framework has been widely explored in the data translation problem in other domains, such as image style transfer in CV and text style transfer in NLP. Compared with these works, although the proposed approach studies a new problem, the major idea is the same as the existing studies. For the graph encoder and graph decoder, they are designed based on the idea of graph neural networks, where some propagation layers are designed to propagate information across different nodes. Although the propagation layers are specifically designed for the graph translation problem, I feel like they are not so different from existing studies (e.g, message passing neural network, graph U-net). Therefore, from the model-wise, this paper combines several existing ideas, but does not provide new insights or techniques, so the contribution is quite limited.

2. The writing can be further improved.
The paper is not very well-written. Some parts of the paper are quite hard to follow, and the intuition behind the approach is not well explained. In section 3.2, it is said that "the approach learns global information by looking for more virtual neighbors regarding the latent relations". Here, it is unclear to me what is a virtual neighbor, and what is a latent relation. The authors try to illustrate their idea in figure 2, but the figure is also quite hard to understand. It would be better if the authors could explain the idea of the encoder in a more intuitive way, or give more concrete examples for illustration. Besides, equation (4) (5) and (7) are also hard to understand. The notations in these equations are quite messy, where multiple indices are used (e.g., i, j, k, l, m, n), and the intuition underlying the equations is not well explained, making it hard to understand how the encoder and the decoder work.
Also, there are many typos in the paper. For example:
In the directed graph, each node have incoming edge(s) and out-going edge(s) -> In the directed graph, each node has incoming edge(s) and out-going edge(s)
in the "node comvolution" layer -> in the "node convolution" layer
First, the “node deconvolution” layer are used to generates -> First, the “node deconvolution” layer is used to generate
The caption of table 1 says that the table shows the node degree distribution distance, but from the main body of texts, only four metrics are about the distribution distance, which is inconsistent to the caption.

Overall, the intuition of the proposed approach is not well explained, and there are many typos to be fixed, so I feel like the writing of the paper should be further improved.


**Experience Assessment:**

I have published in this field for several years.

**Review Assessment: Checking Correctness Of Derivations And Theory:**

I carefully checked the derivations and theory.

**Review Assessment: Checking Correctness Of Experiments:**

I carefully checked the experiments.

**Review Assessment: Thoroughness In Paper Reading:**

I read the paper thoroughly.

---

> ### Author Response · Authors · 2019-11-12
> **Authors' Response to Reviewer #1 (Part I)**
>
> Dear Reviewer,
>
> We thank you for your careful reading and valuable comments. Below we address the concerns mentioned in the review:
> --------------------------------------------
> Q1:  Concerns of the novelty of the proposed approach. Compared with these works in CV and NLP, although the proposed approach studies a new problem, the major idea is the same as the existing studies. For the graph encoder and graph decoder, …, I feel like they are not so different from existing studies (e.g, message passing neural network, graph U-net).
>
> Answer: Generative adversarial network is a celebrated generic learning framework, which has been widely applied in many domains including CV and NLP. For its tens of thousands of followed works (based on google scholar), they all shared the same learning framework or the “main idea” is the same. It is unfair to argue the novelty of our work because GAN is served as our learning framework. Instead, we argued that in order to apply GAN on graph domains, we have designed a very novel conditional GAN framework for graph translation, namely GT-GAN which consists of a novel graph translator and conditional graph discriminator, both of which have been proposed for the first time (to the best of our knowledge).
>
> More specifically, for the graph encoder, most of the graph encoders such as GCN, GGSNN, MPNN, GAT are shared the similar high-level methodology and the main differences are how to design the aggregation functions and how to pass the “messages” through a set of nodes or both a set of edges and nodes. However, our graph encoder is quite different from the existing GNN variants. We first propose the “edge convolution” layers to learn a group of multi-mode relations from the topology of the input graph, which can include both the n-hop connections and the latent relations that are derived from their adjacent edges/relations. And then the “node convolution” layer is used to embed each node representations by aggregating its “virtual neighbors” that related to each latent relations extracted from different the adjacent edges/relations. As shown in Table 4, we performed the ablation study by replacing our graph encoder with others including GCN, DCNN, and Graph U-Nets and showed that our graph encoder consistently performed the best.
>
> For the graph decoder, this is actually the most challenging part. We are the first to propose the graph deconvolution layer, which has not been studied and handled before. The existing graph decoders (VGAE [1], graphite [2]) are all about just calculating the similarity of node embedding to generate the edges. They are over-restrictive by assuming the similar node embeddings indicate the edges between them. Graph U-net [3] is designed just for graph embedding on a fixed graph topology without generating graph edges, which is different from ours.
> In summary, our decoder is the first graph deconvolution method and our encoder is different from existing methods. Our ablation studies (in Sec. 4.5)  have validated the superiority of our proposed graph deconvolution and encoder over the existing GNN-based encoders and decoders.
>
> [1] Kipf, T. N., & Welling, M. (2016). Variational graph auto-encoders. arXiv preprint arXiv:1611.07308.
> [2] Grover, A., Zweig, A., & Ermon, S. (2018). Graphite: Iterative generative modeling of graphs. arXiv preprint arXiv:1803.10459.
>
> --------------------------------------------
> Q2: In section 3.2, it is said that "the approach learns global information by looking for more virtual neighbors regarding the latent relations". Here, it is unclear to me what is a virtual neighbor, and what is a latent relation. It would be better if the authors could explain the idea of the encoder in a more intuitive way, or give more concrete examples for illustration.
>
> Answer: This is a great suggestion. We have followed your suggestions by replacing Fig. 2 and adding a new Fig. 3 in order to make it more clear.
>
> In particular, “Virtual neighbors” refers to the nodes that are linked due to the extracted latent relationships instead of the original edges. For example, assume we have two nodes 1 and 2. Though nodes 1 and 2 are not connected in the input graph, if they have the same node degree, then based on the latent relation of “node degree similarity”, they can be treated to be virtually connected. And if for scale-free graphs, node degree similarity is a very important criterion since it indicates the knowledge that these two nodes have a similar probability to connect to others and that knowledge may be critical in generating the output graph. The rationale why we will extract various latent relationships among nodes before doing node embedding is that we would like to perform node embeddings not only based on the original edges but also on more hidden relations among nodes.

---

> ### Author Response · Authors · 2019-11-12
> **Authors' Response to Reviewer #1 (Part II)**
>
> --------------------------------------------
> Q3: Equation (4) (5) and (7) are also hard to understand. The notations in these equations are quite messy, where multiple indices are used (e.g., i, j, k, l, m, n), and the intuition underlying the equations is not well explained, making it hard to understand how the encoder and the decoder work.
>
> Answer: We tried to provide more details in our convolution and deconvolution and that’s why we decided to keep element-wise matrix operations. In order to further explain how the encoder and decoder works, we now provided a new Figure (Figure 3) to elaborate on the matrix operations in the updated manuscript. We hope these figures are helpful for you to better understand our proposed methods.
>
> --------------------------------------------
>  Q4: there are many typos in the paper.
>
> Answer: Thank you for pointing out these typos. We have corrected them in the updated manuscript.
>
> --------------------------------------------
> Q5: The caption of table 1 says that the table shows the node degree distribution distance, but from the main body of texts, only four metrics are about the distribution distance, which is inconsistent with the caption.
>
> Answer: Yes, this is a mistake. We have updated the table name in the revision.

---

### Official Review · AnonReviewer2 · 2019-10-25
**Official Blind Review #2**

**Rating:** 8

**Review:**

In this work the authors tackle the problem of generating a given graph to a target output graph.  To achieve this they develop a novel deep graph generative model. The authors place a lot of emphasis on scalability. This is indeed a major computational bottleneck in prior work, allowing the deep generative models to generate graphs with few tenths of nodes.   The authors propose an architecture that consists of a graph translator, and a conditional graph discriminator. The GAN approach is able to give significant insights into the conditional distribution p(G|H) where H is the input graph. For the graph translator, the authors design novel graph encoders and decoders. The proposed encoder-decoder achieve the best possible results, compared to using established encoder/decoders as shown in ablation study.  The authors analyze the computational complexity of their work, and while they do not discuss the computational complexities of the other methods, it is clear from the experiments that their method scales better (e.g., Figure 4). To evaluate the output of the architecture, they use a variety of different graph characteristics. The proposed method outperforms the state-of-the-art. Furthermore, the proposed method is able to detect interesting anomalies as illustrated in the appendix  (hacker detection).

Overall this paper makes several interesting contributions on a challenging problem, it is well written (modulo few typos, e.g.,10X instead of 10$\times$, generater->generator), and has convincing experiments. For these reasons I suggest its acceptance.

**Experience Assessment:**

I have read many papers in this area.

**Review Assessment: Checking Correctness Of Derivations And Theory:**

I assessed the sensibility of the derivations and theory.

**Review Assessment: Checking Correctness Of Experiments:**

I carefully checked the experiments.

**Review Assessment: Thoroughness In Paper Reading:**

I read the paper thoroughly.

---

> ### Author Response · Authors · 2019-11-12
> **Reply to Reviewer #2**
>
> Dear Reviewer,
>
> We thank you for the thorough reading and valuable comments. Below we address the concerns mentioned in the review:
>
> ----------------------------------
> 1) Overall this paper makes several interesting contributions to a challenging problem, it is well written (modulo few typos, e.g.,10X instead of 10, generater->generator), and has convincing experiments. For these reasons, I suggest its acceptance.
>
> Reply: We are very grateful to the reviewer for this accurate summary, and for the kind recognition of our key contributions. We have already corrected the typos in our paper. Furthermore, we have also added Fig. 3 and replaced the Fig. 2 to make the operations of the proposed encoder and decoder more clearly.
>
> ----------------------------------
> 2) The authors analyze the computational complexity of their work, and while they do not discuss the computational complexities of the other methods, it is clear from the experiments that their method scales better (e.g., Figure 4).
>
> Reply: Yes, this is a great suggestion although we discussed that scalability is a serious limitation of existing works. We will add more formal discussions of the computational complexity of other works in Sec 3.5.

---

### Decision · Program_Chairs · 2019-12-19

**Decision:**

Reject

**Comment:**

This paper studies a problem of graph translation, which aims at learning a graph translator to translate an input graph to a target graph using adversarial training framework. The reviewers think the problem is interesting. However, the paper needs to improve further in term of novelty and writing.